# Antimicrobial Susceptibility Profiles of Bacteria Commonly Isolated from Farmed Salmonids in Atlantic Canada (2000–2021)

**DOI:** 10.3390/vetsci9040159

**Published:** 2022-03-25

**Authors:** Rasaq Abiola Ojasanya, Ian A. Gardner, David B. Groman, Sonja Saksida, Matthew E. Saab, Krishna Kumar Thakur

**Affiliations:** 1Department of Health Management, Atlantic Veterinary College, University of Prince Edward Island, Charlottetown, PE C1A 4P3, Canada; iagardner@upei.ca (I.A.G.); ssaksida@upei.ca (S.S.); kthakur@upei.ca (K.K.T.); 2Aquatic Diagnostic Services, Atlantic Veterinary College, University of Prince Edward Island, Charlottetown, PE C1A 4P3, Canada; groman@upei.ca (D.B.G.); msaab@upei.ca (M.E.S.)

**Keywords:** antimicrobial resistance, antimicrobial susceptibility, bacterial isolates, salmon, Atlantic Canada, aquaculture

## Abstract

Bacterial infection and antimicrobial resistance are important constraints in the production and sustainability of farmed salmonids. This retrospective study aimed to describe the frequency of bacterial isolates and antimicrobial resistance profiles in salmonid aquaculture in Atlantic Canada. Bacterial isolates and antimicrobial susceptibility testing (AST) results assessed by disk diffusion testing were summarized for 18,776 Atlantic salmon (*Salmo salar*) and rainbow trout (*Oncorhynchus mykiss*) samples from 2291 unique cases submitted to the Atlantic Veterinary College, Aquatic Diagnostic Services Bacteriology Laboratory from 2000 to 2021. Kidney was the most commonly submitted tissue (60.29%, n = 11,320), and these specimens were mostly submitted as swabs (63.68%, n = 11,957). The most prevalent pathogens detected in these cases were *Yersinia ruckeri* type 1 (5.54%, n = 127), *Renibacterium salmoninarum* (2.10%, n = 48), *Aeromonas salmonicida* (atypical) (1.66%, n = 38), and *Pseudomonas fluorescens* (1.22%, n = 28). Most bacterial isolates tested (n = 918) showed resistance to florfenicol, oxytetracycline, ormetoprim-sulfadimethoxine, and trimethoprim-sulfamethoxazole, but not to enrofloxacin. This report provides baseline data for antimicrobial surveillance programs that investigate emerging antimicrobial resistance trends in salmonid aquaculture in Atlantic Canada.

## 1. Introduction

Bacterial diseases are one of the challenges that confront aquaculture worldwide [1]. Environmental stress compounded by high-density rearing conditions increase the susceptibility of farmed fish to bacterial infections, and may contribute to elevated mortality and significant financial losses [2]. Occurrence of bacterial diseases in aquaculture has necessitated use of antimicrobials by fish farmers to mitigate losses [3].

Use of antimicrobials to prevent and treat bacterial diseases, among other factors such as warming waters [4], has advanced concern about antimicrobial resistance in food animal production systems. The emergence of antimicrobial resistance is of serious concern for the sustainability of fish production [3,5]. Several scientific studies have been conducted on antimicrobial resistance in salmonids from marine and freshwater environments [6,7,8]. Of note are studies by Miranda and Zemelman [8], who evaluated the resistance of Gram-negative bacterial isolates recovered from freshwater salmonid aquaculture, to several antimicrobials using agar disk diffusion method, as well as Balta et al. [9] who investigated the resistance of *Yersinia ruckeri* isolates from rainbow trout to nine antimicrobials. 

To our knowledge, there has not been a retrospective laboratory study describing the antimicrobial resistance of major bacterial pathogens of fin-fish in Canadian aquaculture. This study describes the frequency of bacterial isolates and antimicrobial resistance profiles in salmonid samples submitted to the Atlantic Veterinary College, Aquatic Diagnostic Services Bacteriology Laboratory (AVC ADSBL) during the past two decades in which bacterial testing methods have changed or evolved with improvements in technology and so did our understanding of the host, agent, and environmental interactions. 

## 2. Materials and Methods

### 2.1. Data Source 

Records of microbiological analyses of Atlantic salmon (*Salmo salar*) and rainbow trout (*Oncorhynchus mykiss*) samples from 1 January 2000 to 30 April 2021 were analyzed in this study. These samples were submitted to the Atlantic Veterinary College, Aquatic Diagnostic Services Bacteriology Laboratory (AVC ADSBL) from aquaculture facilities in New Brunswick, Nova Scotia, Newfoundland and Labrador, and Prince Edward Island on the Atlantic coast of Canada. The variables provided in the retrieved data included the following: date of sample submission, sample identification number (unique case number assigned by the submitter), species of fish, anatomic site of a sample, type of specimen (e.g., swab, tissue, fluid, or culture plates), bacterial organism isolated, and antimicrobial susceptibility test results. Samples submitted to AVC ADSBL by any aquaculture facility on any given day for bacterial culture and isolation were provided with the same identification number and are referred to as a case for the purpose of the analysis. Specific geographic locations or the province of aquaculture facilities submitting the samples are not reported for confidentiality reasons.

### 2.2. Bacterial Culture, Identification, and Antimicrobial Susceptibility Testing

Bacteria were isolated from swabs, culture plates, tissues, and fluids using standard microbiological techniques [10]. All samples were aseptically plated onto Columbia agar with 5% sheep blood (BA) or onto BA supplemented with 2% sodium chloride for salmonids from marine environments. BA plates were incubated at both 15 °C and 22 °C for at least 7 days. Selective culture media for *Flavobacterium* spp. (Cytophaga agar), *Renibacterium salmoninarum* (selective kidney disease medium), and *Vibrio* spp. (Thiosulfate-citrate-bile salts-sucrose agar) were used as indicated by sample site or clinical history. Bacterial colonies were presumptively identified using Gram stain morphology, catalase and oxidase activity, motility, and a battery of biochemical tests such as carbohydrate oxidation and fermentation. Sensitivity to the vibriostatic agent O129 was also used to further classify Gram-negative isolates. The identity of the significant salmonid pathogens was confirmed using antisera agglutination or immunofluorescent antibody staining techniques. Antimicrobial susceptibility testing (AST) was performed using the disk diffusion assay following the Clinical and Laboratory Standards Institute (CLSI) standard testing methodology for bacteria isolated from terrestrial and aquatic animals. AST for *R. salmoninarum* is usually not performed by AVC ADSBL. For organisms with no standardized methodology, modifications appropriate for the organism were used (e.g., culture medium or temperature). Facultative halophiles were tested on Mueller Hinton Agar (MHA), while obligate halophiles were tested on MHA supplemented with 2% sodium chloride. Fetal calf serum was added to MHA for *F.* spp. that would not grow on MHA. Disk diffusion plates for most isolates were incubated at 22 °C, and rapid-growing organisms were incubated for 48 h. Slow-growing organisms were incubated for 4 days. Zones of inhibition were interpreted based on CLSI guidelines that were current when AST was performed. For organisms where no guidelines exist, interpretative criteria for similar antimicrobial or organism combinations were used. Quality control was performed as recommended by CLSI guidelines.

Antimicrobial susceptibilities to florfenicol, oxytetracycline, ormetoprim-sulfadimethoxine, trimethoprim-sulfamethoxazole, and enrofloxacin were determined. Antimicrobial susceptibilities were interpreted and grouped into three categories: susceptible, resistant, or intermediate (also known as moderately susceptible). The interpretative criteria (e.g., breakpoint zone diameters) were not available for analysis.

### 2.3. Data Management

Retrieved data were checked for consistency and edited whenever necessary to group entries that were entered differently (such as spelled differently, different ways of inputting bacterial species names, etc.). Bacterial names reported are based on current taxonomy and may not be the name reported in the database. Entries with an identical fish identifier, sample identifier, anatomic site, bacterial isolate, and antimicrobial susceptibility profiles were deemed duplicates and were excluded from the analyses upon verification as duplicate entries by the database curator. Entries with no record of bacterial isolate detection, with contaminants as the only entry, or with the growth of other organisms (such as fungi and algae) were collectively grouped and removed from analysis. Entries with clearly-identified sample sites, such as kidney and skin, were maintained separately. Entries with sample sites without clear identification, such as culture, lesions, sample, and unspecified, were collectively grouped as “unspecified”. Entries with samples from abdominal cavity, blood, bladder, brain, ova, eye, fin, gill, head, heart, intestine, jaw, liver, mouth, muscle, ovarian fluid, peritoneal cavity, spleen, tail, vent, wound, and yolk sac were collectively grouped as “others”.

### 2.4. Statistical Analysis

The frequencies of annual and monthly cases of submissions, overall samples (by sample sites) submitted, the annual distribution of sample submitted from different sites, bacterial isolates recovered from these sites, unique cases of bacterial isolates detected in the two species of salmonid, and annual frequency of major bacterial organisms detected from salmonid cases submitted were summarized using Stata version 15 (StataCorp, College Station, TX, USA, 2017). Bacterial isolates recovered from different sample sites were presented as percentages of the total submitted samples with 95% confidence intervals calculated using the exact method.

The antimicrobial susceptibility of individual isolates was recorded as susceptible (S), intermediate (I), or resistant (R) in the database. Descriptive statistics were used to summarize frequencies of bacterial isolates with antimicrobial susceptibility test results as percentages of S, I, and R.

To visualize the annual antimicrobial resistance trends and pattern for the most frequently tested bacterial isolates (and those that were identified to the species-level) for antimicrobial susceptibility, the data on annual percentage of resistance for *Aeromonas salmonicida* (atypical), *Aeromonas salmonicida* (typical), *Pseudomonas fluorescens*, and *Yersinia ruckeri* type 1 were extracted for each antimicrobial drug tested in this study and were represented with heat maps using pheatmap package [11] in R Statistical Software (version 1.4.17; R Core Team, 2019).

Logistic regression was used to assess the temporal trend in antimicrobial resistance profiles across study years. The resistant and intermediate results for each of the most frequently tested bacteria (and those that were identified to the species-level) were grouped as resistant (R = 1) compared with isolates classified as susceptible (R = 0). For the most frequently tested bacteria and antimicrobial combinations, the probability of antimicrobial resistance (R = 1) was modeled as the binary outcome with study year as a continuous explanatory variable. The assumption of linearity, however, could not be met, and study years were finally modeled as categorical variables to assess inter-annual differences in resistance profiles. The study years were also dichotomized (as decades, up to 2010 and after) and were explored as a factor in these models. Clustering of samples within the submitted cases was accounted for by using a mixed effects logistic regression with cases as the random effect. Model fit was assessed with the Hosmer–Lemeshow goodness-of-fit, and the Wald test was used to infer significant differences (*p* < 0.05) in resistance profiles.

## 3. Results

Approximately 90% of the samples submitted were from Atlantic salmon (n = 16,868), while 10% were from rainbow trout (n = 1908). These samples corresponded to 2291 unique cases (defined as group of samples submitted from a unique facility on any given day for bacterial culture and isolation) recorded in the database for this study period that met the inclusion criteria. Each case included sample submissions from one to many fishes (mean = 8.20, median = 5.00, range = 1.00–219.00). The average number of unique cases investigated by the AVC ADSBL per year and month was 104.00 and 8.20, respectively. The annual and monthly frequencies of submitted cases are presented in Figure 1 and Figure 2, respectively. AVC ADSBL received most cases, in the months of April, August, and from October to December. The annual distributions of anatomic sites from which the submitted samples originated are presented in Appendix A. Kidney samples were the most commonly submitted anatomic site for bacterial isolation and culture (60.28%, n = 11,320), followed by samples that had no clear identification in the database and were collectively grouped as “unspecified” (20.79%, n= 3905). In comparison, skin was the least frequently submitted sample (8.07%, n = 1517). The frequencies of anatomic sites from which submitted samples originated are presented in Appendix A. Specimens for bacterial investigation were mostly submitted as swabs (63.68%, n = 11,957), followed by culture plates (28.11%, n = 5278), tissues (8.01%, n = 1505), and fluids (0.19%, n = 36), as presented in Appendix A.

Of the 2291 cases submitted for culture and isolation, 22.87% (n = 524) had no bacterial growth from any of the submitted specimens, while 77.12% (n = 1767) yielded bacterial growth from at least one specimen. The level to which bacteria were identified and the frequency of cases which had AST are presented in Figure 3. The mean and median numbers of bacteria identified to species-level from a case were 1.10 and 1.00, respectively. In 5.0% of the cases, however, two or more bacteria from a case were identified to species-level (or had co-infection).

Frequencies of bacterial isolates detected in the cases of salmonid species that were identified up to species-level are presented in Table 1. *Y. ruckeri* type 1, *R. salmoninarum*, *A. salmonicida* (atypical), and *P. fluorescens* in descending order were the most prevalent pathogens detected in Atlantic salmon. In rainbow trout, *R. salmoninarum*, *F. psychrophilum*, *Aeromonas hydrophila*, and *Aeromonas sobria*, were the most common pathogens identified. *Y. ruckeri* type 1, *A. salmonicida* (atypical), and *A. salmonicida* (typical) were not detected in any submitted rainbow trout cases. The annual frequency of the most prevalent pathogens detected from salmonid cases submitted to AVC ADSBL is presented in Figure 4. Overall, there were decreasing temporal trends for the detection of *Y. ruckeri* type 1, *R. salmoninarum*, *A. salmonicida* (atypical), and *A. salmonicida* (typical). *Y. ruckeri* type 1 was mostly detected during the study period. Co-infections in this study mostly involved *R. salmoninarum* and *Y. ruckeri* type 1, and *R. salmoninarum* and *A. salmonicida* (atypical).

Frequencies of bacterial isolates detected from anatomic sites of salmonid samples submitted to the AVC ADSBL, irrespective of the availability of their antimicrobial susceptibility records, are presented in Table 2. *Y. ruckeri* type 1 was the most common bacterial species isolated from the kidney (7.70%) and unspecified (15.00%) anatomic sites. *F. columnare* was more commonly isolated from the skin (4.60%), while *P.*
*fluorescens* mainly was isolated from “others” (2.60%). Bacterial isolates that were not identified to genus and species-level and those that were less frequently isolated from these anatomic sites are presented in Appendix A.

Of the 557 cases which had AST, 62.29% (n = 347) had a single bacterial isolate tested for antimicrobial susceptibility, while 37.70% (n = 210) had two or more bacterial isolates tested for antimicrobial susceptibility. One hundred and seventy cases had two or more AST done for the same bacterial isolate. The number of isolates tested per case ranged from one to nine (all nine isolates tested were from a case in which *A. salmonicida* (atypical) and *Y. ruckeri* type 1 were detected and these isolates had similar profile of antimicrobial susceptibility) with the mean and the median of isolates tested per case of 1.60 and 1.00, respectively. Antimicrobial susceptibilities for bacterial isolates identified to species-level and those identified to genus-level are summarized in Table 3 and Appendix A, respectively. Overall, 918 bacterial isolates had antimicrobial susceptibility test results available. *Y. ruckeri* type 1 (n = 215), *A. salmonicida* (atypical) (n = 111), *P. flourescens* (n = 47), and *A. salmonicida* (typical) (n = 29) were the most frequently tested bacterial isolates (that were identified to the species-level) in this study. *A. salmonicida* (atypical) was highly resistant (>90%) to oxytetracycline. None of the bacterial isolates with AST showed resistance to enrofloxacin during the study period.

Only 2.50% (n = 23) of the total bacterial isolates which had AST were resistant to all five antimicrobials evaluated. *A. salmonicida* (atypical) (n = 11), which originated from a total of six cases from three different years (2000, 2008, and 2009), was resistant to most of the antimicrobials tested, followed by *P. fluorescens* (n = 5) from a total of five cases from three different years (2000, 2006, and 2009). About 3.80% (n = 35) of the total bacterial isolates with AST were resistant to four antimicrobials tested, while 9.60% (n = 88) were resistant to three antimicrobials.

The temporal antimicrobial resistance patterns for the most frequently tested bacterial isolates (identified to species-level) are presented in Figure 5. For bacteria where both ormetoprim-sulfadimethoxine (Romet-30^®^) and trimethoprim-sulfamethoxazole were on the panel, 94.20% (405 out of 431) had a similar susceptibility/resistance profiles to these antimicrobials.

Overall, significant inter-annual differences in resistance were only evident among *A. salmonicida* (atypical) isolates tested for trimethoprim-sulfamethoxazole after accounting for the clustering of samples within the cases. None of the bacteria tested had an increasing temporal resistance trend (for the years that data was available) for any of the tested antimicrobials.

## 4. Discussion

Findings from the present study indicate a low frequency of bacterial isolation among submitted salmonid samples to AVC ADSBL and limited evidence of resistance in the antimicrobial susceptibility profile of tested organisms. AVC ADSBL conducts a substantial amount of all antimicrobial susceptibility testing of fish isolates in the Atlantic Provinces of Canada. Scientific reports on the most frequently isolated fin-fish bacterial pathogens, antimicrobial use, and trends in antimicrobial resistance are essential to policymakers, aquaculture veterinarians, fish farmers, and the public to support decision-making related to antimicrobial treatments. The current study was motivated by the fact that empirical data on antimicrobial susceptibility for the treatment of bacterial disease in aquaculture are not publicly available from Atlantic Canadian aquaculture operations. This study provides information on trends of potentially important bacterial pathogens in Atlantic Canada and their resistance profile that can assist aquaculture veterinarians, especially those in Atlantic Canada, in making rational decisions on bacterial pathogen antimicrobial use.

*Y. ruckeri* type 1, *R. salmoninarum*, *A. salmonicida* (atypical), and *P. fluorescens* were the most frequently detected bacterial species from the submitted cases. These pathogens had been previously reported in farmed salmonid [8,12,13,14]. In aquaculture, more severe disease conditions are mostly caused by Gram-negative bacteria such as *A.*
*hydrophila*, *A.*
*salmonicida*, *F.*
*psychrophilum*, *Y.*
*ruckeri*, *Vibrio*
*anguillarum*, *Vibrio*
*harveyi*, *Tenacibaculum maritimum*, *Moritella viscosa*, *Piscirickettsia salmonis*, *P.*
*fluorescens*, *Edwardsiella piscicida*, and *Citrobacter freundii* [15,16]. Gram-negative bacteria were more commonly detected in this study than Gram-positive isolates, which is in agreement with the report of Lewbart [17], who observed that bacterial infections in farmed fish are primarily associated with Gram-negative organisms. *Y. ruckeri* type 1 was detected in several anatomic sites in cases of Atlantic salmon but was not identified in any rainbow trout samples, even though trout are considered more susceptible to this infection [18]. *A. salmonicida* (atypical) was also not detected in any rainbow trout samples and was detected only at low frequency in Atlantic salmon samples. There were decreasing temporal trends for the detection of these four pathogens in this study. This likely reflects continued strengthening of health management procedures and the effectiveness of biosecurity measures in salmonid aquaculture in Atlantic Canada [19,20].

This is the first study from Canada to report comprehensive data on antimicrobial susceptibility patterns of major bacterial pathogens of salmonids. A total of 918 cultured bacterial isolates had AST results based on veterinarians’ request at the point of sample submission to the AVC ADSBL. Antimicrobial treatments in Canadian aquaculture require veterinary prescription and are usually provided in the form of a medicated feed. Florfenicol, oxytetracycline, ormetoprim-sulfadimethoxine, and trimethoprim-sulfamethoxazole are antimicrobials authorized for use in aquatic food animal production in Canada [21]. In addition to these approved antimicrobials, enrofloxacin was included in the sensitivity panel used by AVC ADSBL for all cultured bacterial isolates that were tested. This antimicrobial is widely used in both human and animal health and may have been used by fish health veterinarians at some point and added to the panel and remained on the panel [22,23]. Furthermore, extra-label use of this drug has been effective, especially in the treatment of brood-stock and other valuable aquatic species at risk [24,25].

Resistance to all tested antimicrobials was detected in five cases of *P. fluorescens* in 3 years and six cases of *A. salmonicida* (atypical) in 3 years in this study. There is no indication that this resistance is innate or acquired. However, it has been reported that antimicrobial resistance in *P. fluorescens* could be due to the presence of multi-drug efflux systems which is responsible for the low permeability of the outer membrane of most Gram-negative bacteria [26]. However, *P.* spp. infection is not a problem in aquaculture as it is known as an indicator organism that grows due to dissolved oxygen depletion, thereby reflecting water quality and it is not normally treated for [27]. *P. fluorescens* was highly susceptible to enrofloxacin, suggesting that enrofloxacin can be effective as an anti-pseudomonal drug (extra-label use) in aquaculture in Canada though quinolones (enrofloxacin is a fluoroquinolone) are listed by Health Canada and WHO as a category 1 antimicrobial agent for human use and caution should be used when considering its use in food animals [28,29,30,31].

Antimicrobial resistance in *A. salmonicida* (atypical) could be attributable to the presence of R-factor plasmid which has been reported in several strains of *A. salmonicida* in salmonids, tilapia, and carps [32,33,34]. None of the aquatic pathogens isolated and tested in this study had resistance to enrofloxacin. Several studies from Thailand, Chile, and China have reported a low level of resistance to enrofloxacin in aquaculture [8,35,36]. *Y. ruckeri* type 1 had the least resistance to all antimicrobial tested in this study. *Y. ruckeri* is known to lack beta-lactamase enzyme, which increases its susceptibility to beta-lactam antimicrobials. Moreover, high susceptibility has been reported with florfenicol, oxytetracycline, and trimethoprim-sulfamethoxazole [37,38].

Ormetoprim-sulfadimethoxine (Romet-30^®^) and trimethoprim-sulfamethoxazole are sometimes tested simultaneously alongside other antimicrobials in the AVC ADSBL disk diffusion panel to provide invitro data for the potentiated sulfonamides. Due to similarities in the mechanism of action, it was not surprising that, a majority of the time, the sensitivity patterns for the two drugs were found similar [39]. In contrast, both of these antimicrobials were not tested on the same bacterial isolate in other instances. None of the bacteria tested had an increasing temporal trend for resistance for any of the tested antimicrobials over the 21-year time period. Only *A. salmonicida* (atypical) (mainly detected in the last decade) had significant inter-annual differences in trimethoprim-sulfamethoxazole resistance. Since *Y. ruckeri* type 1 was not resistant to five of the antimicrobials tested (florfenicol, ormetoprim-sulfadimethoxine, trimethoprim-sulfamethoxazole, and enrofloxacin), it could not be modeled for inter-annual resistance variability.

Early detection of bacterial diseases in aquaculture can support salmonid aquaculture through timely intervention [40]. Many Gram-negative organisms that affect salmonids present similar clinical signs such as systemic septicemia, localized septic lesions, gill disease, ulcerative skin conditions, and gross lesions at post-mortem examination, necessitating additional tools to be used to come up with a diagnosis [10]. Therefore, bacterial culture, isolation, and AST are important in diagnosing and treating these infections [41]. However, a Gram-positive bacterium, such as *R. salmoninarum*, is a slow-growing organism; consequently, it takes extended time (6 weeks to 12 weeks) for this pathogen to grow in the laboratory and conduct AST [42]. Bacterial kidney disease in salmonids, caused by *R. salmoninarum*, has pathognomonic lesions such as systemic pyogranuloma with necrotic foci in most visceral organs, especially in the kidneys [43]. As such, veterinarians do not necessarily need to rely on bacterial culture and isolation to diagnose bacterial kidney disease and would have already started the treatment regimen by the time they receive bacteriology results. In this study, *R. salmoninarum* was diagnosed with immunofluorescence antibody test (IFAT) using AVC ADSBL made monoclonal antibodies. Kidney tissues and swabs or reproductive fluids were cultured on selective kidney disease medium (SKDM) agar plates and cultures on SKDM were confirmed using IFAT.

Atlantic salmon samples were more commonly submitted throughout the study than samples from rainbow trout, which is consistent with Statistics Canada [44] data showing Atlantic salmon production is larger than trout in Canada. The highest number of cases submitted to AVC ADSBL by aquatic facilities was between 2008 and 2013. Still, most of the samples submitted for bacterial culture and isolation between these periods yielded no bacterial growth. Most of the bacteria cultured were not identified up to the species and were only differentiated as mixed microbiota or Gram-negative bacilli. AVC ADSBL received most cases in April, August, and from October to December, which is a period when salmonids are transferred from hatcheries to salt water sites and most samples submitted during the time could be for screening purpose. Most submitted specimens were from the kidney and these were predominantly submitted as swabs. Kidney tissue retains infectious bacteria longer than any other organs and it is one of the most preferred anatomic sites used for lethal sampling in fish [45].

Some limitations of the study include not being able to differentiate if the multiple specimens submitted for a particular case belonged to the same fish or multiple fish. Neither the age of fish from which samples were collected nor the type of facility submitting the sample was known. In addition, most bacteria recovered in the study were not identified to the species-level. For instance, some were only classified as mixed microbiota, Gram-negative bacilli, Gram-positive bacilli, Gram-positive bacteria, etc., which did not provide insights from clinical or epidemiological perspectives. Several cultured bacteria were also not identified to species-level, and due to this limitation, the antimicrobial sensitivity profiles for many isolates were not summarized and evaluated in the main text but were provided as Appendix A. Our data also did not allow us to differentiate between natural and acquired resistance of the reported bacterial species to tested antimicrobials and, as such, it is likely that some of the observed resistances may be associated with natural resistance. Furthermore, the minimum inhibitory concentrations (MICs) of selected antimicrobials were not determined for the bacterial isolates tested in this study because the instrumentation and aquaculture-specific testing panels for antimicrobials was not available in AVC ADSBL until recently. Due to this limitation, some of the antimicrobial sensitivity trends may not have been captured as accurately as would have been based on MICs. The limitations and gaps identified in this study will guide diagnostic labs improve quality data management and protocols used for diagnostics.

While we acknowledge the above limitations due to the retrospective nature of the study utilizing laboratory database, this study summarized historic information on frequently isolated bacterial pathogens, and their antimicrobial resistance profiles in farmed salmonid in Atlantic Canada. These baseline data provide a benchmark for future surveillance studies of antimicrobial resistance in salmonid aquaculture in Atlantic Canada.

## Figures and Tables

**Figure 1 vetsci-09-00159-f001:**
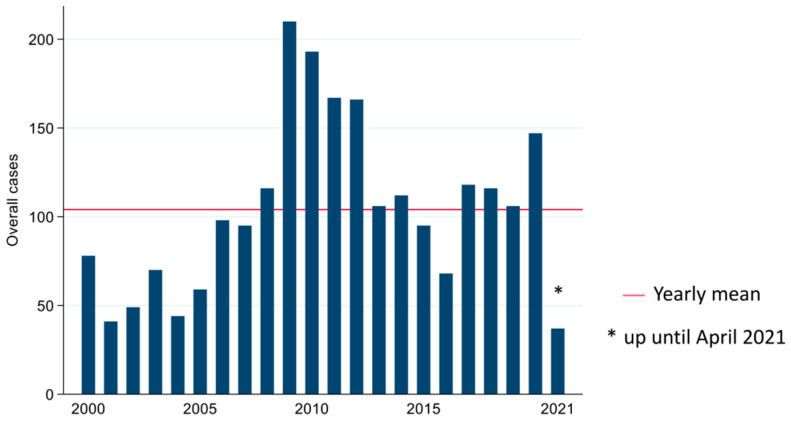
Annual frequency of cases submitted to the AVC ADSBL (2000–2021) for bacterial culture and isolation.

**Figure 2 vetsci-09-00159-f002:**
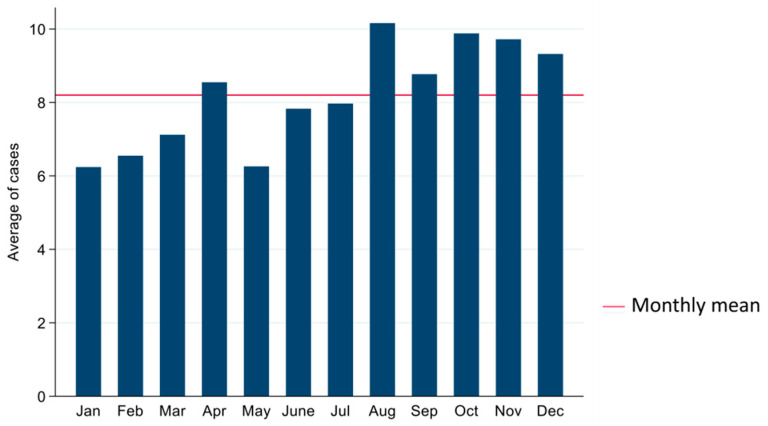
Monthly frequency of cases submitted to the AVC ADSBL from 2000 to 2021 (up until April) for bacterial culture and isolation.

**Figure 3 vetsci-09-00159-f003:**
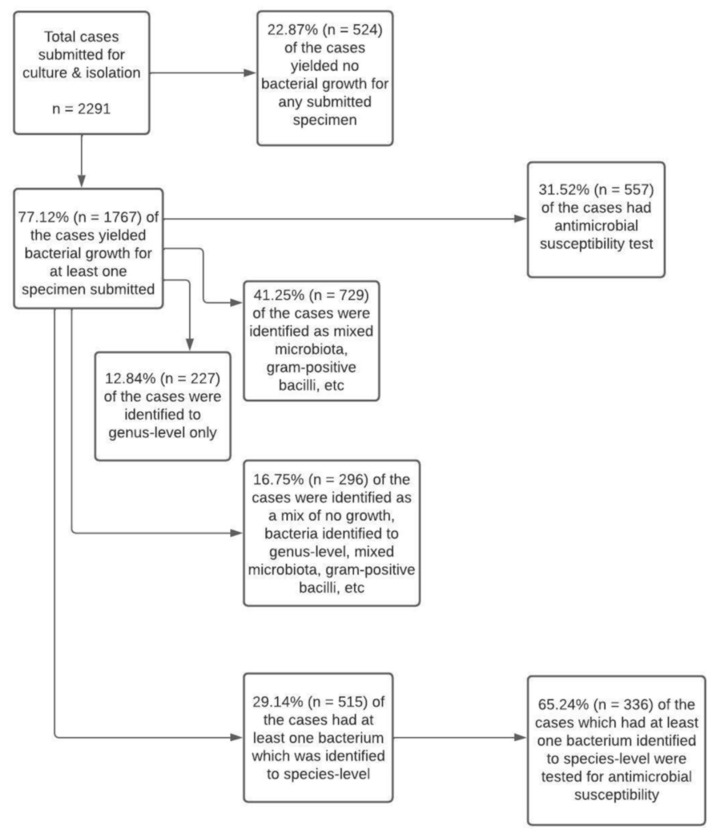
Flowchart summarizing the extent to which salmonid pathogens were identified and cases which had antimicrobial susceptibility test results.

**Figure 4 vetsci-09-00159-f004:**
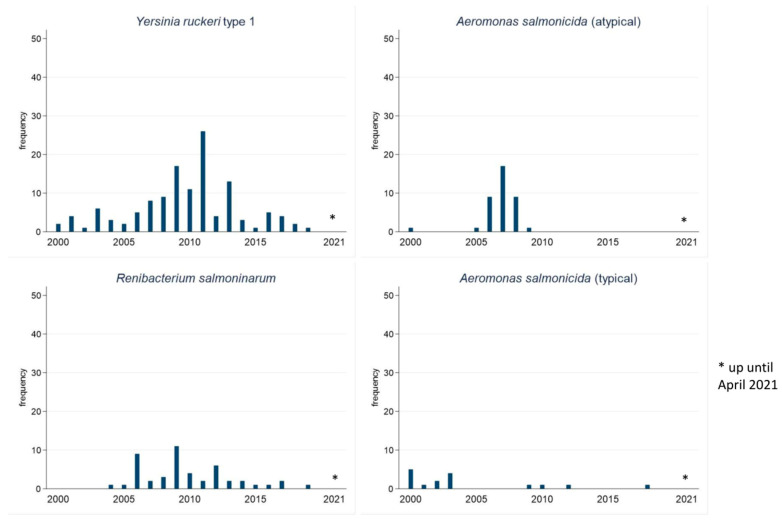
Annual frequency of *Yersinia ruckeri* type 1, *Renibacterium salmoninarum*, *Aeromonas salmonicida* (typical), and *Aeromonas salmonicida* (atypical) detected from salmonid cases submitted to AVC ADSBL (2000–2021).

**Figure 5 vetsci-09-00159-f005:**
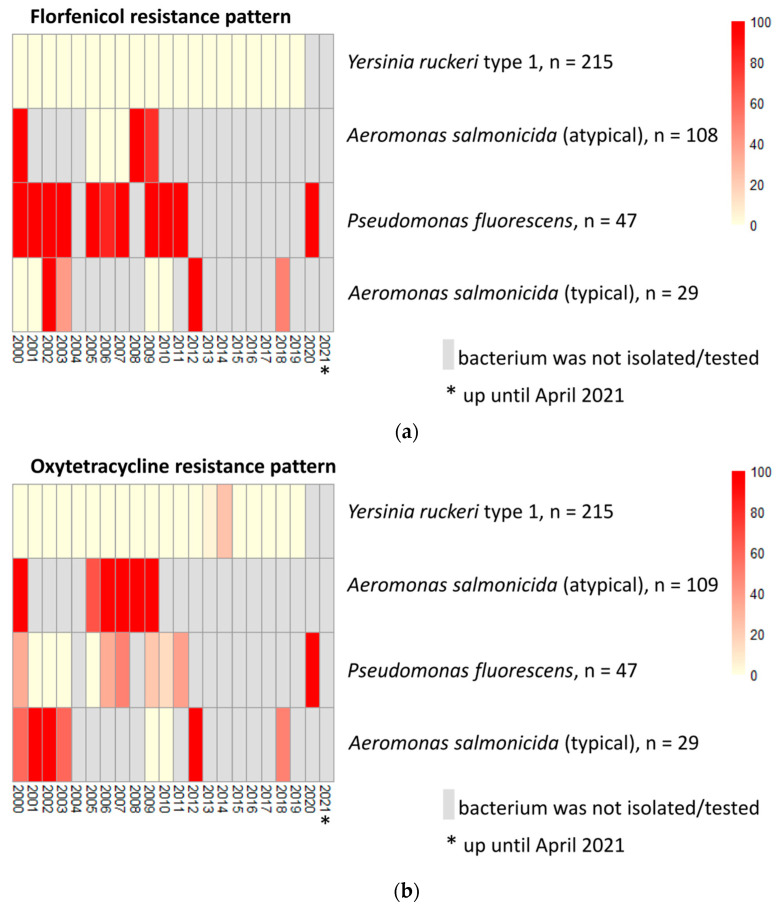
Heat maps show annual antimicrobial resistance trends for (**a**) florfenicol, (**b**) oxytetracycline, (**c**) ormetoprim-sulfadimethoxine, and (**d**) trimethoprim-sulfamethoxazole for the most frequently tested bacterial isolates from salmonid samples (2000–2021).

**Table 1 vetsci-09-00159-t001:** Frequency of bacterial isolates detected in submitted cases to the AVC ADSBL by species of salmonid samples (2000–2021).

Bacterial Isolates	Frequency of Cases (n)	Atlantic Salmon n (%)	Rainbow Trout n (%)
*Yersinia ruckeri* type 1	127	127 (100.00)	0
*Renibacterium salmoninarum*	48	41 (85.40)	7 (14.60)
*Aeromonas salmonicida* (atypical)	38	38 (100.00)	0
*Pseudomonas fluorescens*	28	27 (96.40)	1 (3.60)
*Aeromonas salmonicida* (typical)	16	16 (100.00)	0
*Edwardsiella *piscicida**	14	13 (92.90)	1 (7.10)
*Flavobacterium columnare*	12	11 (91.70)	1 (8.30)
*Aliivibrio salmonicida*	9	8 (88.90)	1 (11.10)
*Aeromonas sobria*	9	7 (77.80)	2 (22.20)
*Aeromonas hydrophila*	8	6 (75.00)	2 (25.00)
*Flavobacterium psychrophilum*	7	2 (28.60)	5 (71.40)
*Vibrio anguillarum* type 1	6	5 (83.30)	1 (16.70)

**Table 2 vetsci-09-00159-t002:** Bacterial isolates detected from different anatomic sites of salmonids samples submitted to the AVC ADSBL (2000–2021).

	Anatomic Sites % (95% CI)			
Bacterial Isolates	Kidney (n = 11,320)	Skin (n = 1517)	Unspecified (n = 3905)	Others (n = 2034)
*Yersinia ruckeri* type 1	7.70 (7.20–8.20)	0.50 (0.20–1.00)	15.00 (13.90–16.20)	1.70 (1.20–2.40)
*Renibacterium salmoninarum*	2.40 (2.10–2.70)	0.00 (0.00–0.20)	1.80 (1.40–2.30)	0.00 (0.00–0.20)
*Aeromonas salmonicida* (atypical)	6.70 (6.20–7.10)	2.00 (1.30–2.80)	2.40 (1.90–2.90)	0.70 (0.40–1.10)
*Pseudomonas fluorescens*	0.90 (0.70–1.10)	2.70 (1.90–3.60)	3.70 (3.10–4.30)	2.60 (2.00–3.40)
*Aeromonas salmonicida* (typical)	0.70 (0.50–0.80)	0.00 (0.00–0.20)	3.10 (2.60–3.70)	0.00 (0.00–0.20)
*Flavobacterium columnare*	0.00 (0.00–3.30 × 10^−2^)	4.60 (3.60–5.80)	0.40 (0.20–0.70)	2.20 (1.60–2.90)
*Flavobacterium psychrophilum*	0.60 (0.40–0.70)	2.70 (1.90–3.60)	0.10 (0.00–0.30)	0.90 (0.60–1.50)

n = number of isolates; CI = confidence interval (estimated using exact method); Unspecified = sample sites without clear identification; Others = include sample sites from the abdominal cavity, blood, bladder, brain, ova, eye, fin, gill, head, heart, intestine, jaw, liver, mouth, muscle, ovarian fluid, peritoneal cavity, spleen, tail, vent, wound, and yolk sac.

**Table 3 vetsci-09-00159-t003:** Antimicrobial susceptibility profiles (% S, I, and R) for tested antimicrobials in bacterial isolates from samples of salmonids (2000–2021).

		Florfenicol				Oxytetracycline				Ormetoprim-Sulfadimethoxine					Trimethoprim-Sulfamethoxazole				Enrofloxacin			
Bacterial Isolates	N	n	S	I	R	n	S	I	R	n	S	I	R	N	n	S	I	R	n	S	I	R
*Yersinia ruckeri* type 1	215	215	100.00	0.00	0.00	215	99.10	0.00	0.90	89	41.40	0.00	0.00	215	215	100.00	0.00	0.00	215	100.00	0.00	0.00
*Aeromonas salmonicida* (atypical)	111	108	85.60	0.00	11.70	109	2.70	0.00	95.50	104	82.90	0.00	10.80	111	109	85.60	0.00	12.60	109	98.20	0.00	0.00
*Pseudomonas fluorescens*	47	47	2.10	0.00	97.90	47	72.40	2.10	25.50	25	0.00	0.00	53.20	47	47	21.30	6.40	72.30	45	95.70	4.30	0.00
*Aeromonas salmonicida* (typical)	29	29	72.40	0.00	27.60	29	41.40	0.00	58.60	23	65.50	0.00	13.80	29	29	72.40	3.50	24.10	29	100.00	0.00	0.00
*Edwardsiella piscicida*	24	24	100.00	0.00	0.00	24	75.00	0.00	25.00	1	4.20	0.00	0.00	24	24	100.00	0.00	0.00	24	100.00	0.00	0.00
*Flavobacterium columnare*	20	20	100.00	0.00	0.00	20	90.00	5.00	5.00	12	5.00	0.00	55.00	20	20	55.00	0.00	45.00	20	100.00	0.00	0.00
*Aliivibrio salmonicida*	11	11	100.00	0.00	0.00	11	100.00	0.00	0.00	11	63.60	0.00	36.40	11	11	63.60	0.00	36.40	11	100.00	0.00	0.00
*Vibrio anguillarum* type 1	10	10	100.00	0.00	0.00	10	100.00	0.00	0.00	10	100.00	0.00	0.00	10	10	100.00	0.00	0.00	10	100.00	0.00	0.00
*Aeromonas hydrophila*	9	9	100.00	0.00	0.00	9	77.80	0.00	22.20	4	44.40	0.00	0.00	9	9	100.00	0.00	0.00	9	100.00	0.00	0.00
*Flavobacterium psychrophilum*	8	8	87.50	0.00	12.50	8	100.00	0.00	0.00	0	0.00	0.00	0.00	8	8	75.00	0.00	25.00	8	87.50	0.00	12.50
*Vibrio anguillarum* type 2	7	7	100.00	0.00	0.00	7	100.00	0.00	0.00	4	57.10	0.00	0.00	7	7	100.00	0.00	0.00	7	100.00	0.00	0.00

N = Total number of isolates, n = number of isolates that were tested for antimicrobial susceptibility, S—Susceptible, I—Intermediate, R—Resistant; cumulative percentages may not add up to 100 as not all isolates had susceptibility test.

## Data Availability

The data presented in this study are available on request from the corresponding author.

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
