# Peer review of "Antimicrobial Susceptibility Profiles of Bacteria Commonly Isolated from Farmed Salmonids in Atlantic Canada (2000–2021)"

_vetsci, 2022, doi:10.3390/vetsci9040159_

Round 1
Reviewer 1 Report
The paper entitled: “Antimicrobial susceptibility profiles of bacteria commonly isolated from farmed salmonids in Atlantic Canada (2000-2021)” is very interesting because it is made over 21 years from a general picture of the presence of antibiotic-resistant bacteria that are commonly isolated in salmonids. The paper is well written and I recommend it for publication.
Author Response
Thanks a lot! I am happy to receive this outstanding feedback on the manuscript.
Reviewer 2 Report
Figure 2. “16.75% of the cases were identified as mix of no growth, bacteria…” is under “77.12% of the cases yielded bacterial growth for at least one specimen submitted”. If it was no growth, how could it be grouped under the 77.12% of cases that yielded bacterial growth? It is confusing.
Author Response
Reviewer 2
Comments and Suggestions for Authors
Figure 2. “16.75% of the cases were identified as mix of no growth, bacteria…” is under “77.12% of the cases yielded bacterial growth for at least one specimen submitted”. If it was no growth, how could it be grouped under the 77.12% of cases that yielded bacterial growth? It is confusing.
Thank you very much for your observation.
For this group, there were multiple specimens that were submitted per case for bacterial culture. In these cases, no growth and bacterial growth were observed together, however, the bacteria that were isolated were only identified either up to the genus-level, mixed microbiota, or as gram-positive bacilli, etc.
Reviewer 3 Report
The article is very interesting and a lot of data was analyzed during 21 years. Unfortunately, some important information was lost, others could not be included in the analysis, but even so, the number of animals examined is impressive. In the Abstract, Introduction and Discussion I have nothing to add, but in Material and Methods some points need to be clarified.
- Which organs were analyzed in each fish that actually were analyzed? In Material and Methods on lines 111 to 114 the "other" organs are listed, but not the ones that were analyzed. This was only made clear in Table 2.
How were these organs collected? Were these fish all adults? Females, Males? Were they alive or dead before collection? If they were alive how were they euthanized? If were dead how They were transported?
- In Figure 2 are the data grouped by months of the different years? For example Jan 2000 to 2021, Feb 2000 to Feb 2021 etc?
- Was 2010 the year that had the most fish samples or the greatest number of species of bacteria detected?
- line 351 - change R. salmoninarum to R. salmoninarum in italics
Author Response
Reviewer 3
Comments and Suggestions for Authors
The article is very interesting and a lot of data was analyzed during 21 years. Unfortunately, some important information was lost, others could not be included in the analysis, but even so, the number of animals examined is impressive. In the Abstract, Introduction and Discussion I have nothing to add, but in Material and Methods some points need to be clarified.
- Which organs were analyzed in each fish that actually were analyzed? In Material and Methods on lines 111 to 114 the "other" organs are listed, but not the ones that were analyzed. This was only made clear in Table 2.
Thank you for this comment. I have provided a supplementary table (Table S1.) showing the frequency of anatomic sites of from which samples/swabs were taken and submitted to the lab.
How were these organs collected? Were these fish all adults? Females, Males? Were they alive or dead before collection? If they were alive how were they euthanized? If were dead how They were transported?
The data on age, sex, live or dead samples, and transport media were not captured in AVC ADSBL database during the study period.
- In Figure 2 are the data grouped by months of the different years? For example Jan 2000 to 2021, Feb 2000 to Feb 2021 etc?
Yes. The data was grouped by months of different years to explore the seasonal patterns in submission of salmonid cases.
- Was 2010 the year that had the most fish samples or the greatest number of species of bacteria detected?
No. 2010 was the year that AVC ADSBL received the highest number of salmonid cases for bacterial culture.
- line 351 - change R. salmoninarum to R. salmoninarum in italics
Thank you. We have corrected that.